# A Novel Shadowgraphic Inline Measurement Technique for Image-Based Crystal Size Distribution Analysis

**Dominic Wirz [1], Marc Hofmann [1], Heike Lorenz [2], Hans-Jörg Bart [1], Andreas Seidel-Morgenstern [2] and Erik Temmel [2,3,*]**

[1] Chair of Separation Science and Technology, TU Kaiserslautern, Gottlieb-Daimler-Straße, 67663 Kaiserslautern, Germany; Dominic.Wirz@mv.uni-kl.de (D.W.); m_hofman@rhrk.uni-kl.de (M.H.); bart@mv.uni-kl.de (H.-J.B.)

[2] Max Planck Institute for Dynamics of Complex Technical Systems, Sandtorstraße 1, 39106 Magdeburg, Germany; lorenz@mpi-magdeburg.mpg.de (H.L.); seidel@mpi-magdeburg.mpg.de (A.S.-M.)

[3] Sulzer Chemtech Ltd., Gewerbestraße 28, 4123 Allschwil, Switzerland

\* Correspondence: erik.temmel@sulzer.com or temmel@mpi-magdeburg.mpg.de

**Abstract:** A novel shadowgraphic inline probe to measure crystal size distributions (CSD), based on acquired greyscale images, is evaluated in terms of elevated temperatures and fragile crystals, and compared to well-established, alternative online and offline measurement techniques, i.e., sieving analysis and online microscopy. Additionally, the operation limits, with respect to temperature, supersaturation, suspension, and optical density, are investigated. Two different substance systems, potassium dihydrogen phosphate (prisms) and thiamine hydrochloride (needles), are crystallized for this purpose at 25 L scale. Crystal phases of the well-known $KH_2PO_4/H_2O$ system are measured continuously by the inline probe and in a bypass by the online microscope during cooling crystallizations. Both measurement techniques show similar results with respect to the crystal size distribution, except for higher temperatures, where the bypass variant tends to fail due to blockage. Thiamine hydrochloride, a substance forming long and fragile needles in aqueous solutions, is solidified with an anti-solvent crystallization with ethanol. The novel inline probe could identify a new field of application for image-based crystal size distribution measurements, with respect to difficult particle shapes (needles) and elevated temperatures, which cannot be evaluated with common techniques.

**Keywords:** optical measurement techniques; crystal size measurement; inline probe; crystal needles

## 1. Introduction

Crystallization is widely applied in agricultural, pharmaceutical, or chemical industry, with an enormous variety of duties and products. Besides purification and concentration of substances, crystallization is mainly applied to produce a particulate phase exhibiting defined properties. The particular requirements can be manifold. While for fine chemicals or active pharmaceutical ingredients (API) the solid-state form is commonly of major interest, a certain crystal size distribution (CSD) and crystal shape is usually demanded. Any of these particular properties affect the product functionality [1], as well as other processes in the downstream procedure (filtration, drying, etc.) [2,3]. Hence, control of the crystal size and solid-state form (e.g., polymorphism) is one important challenge that spreads over all fields of industrial crystallization [4].

Different options like cooling, adding an antisolvent, or evaporation of the solvent can be applied to create the corresponding driving force for the crystallization process. In order to compensate the supersaturation, a solid phase is formed with a specific shape through nucleation or growth, depending on the thermodynamic state of the system. However, crystallization kinetics are complex, nonlinearly connected to the driving force, and highly affected by initial conditions, process disturbances and hydrodynamic effects, which appear during operation.

Beside these fundamental aspects of the solid phase formation, the operation mode of a crystallization process plays a major role. In pharmaceutical industry the vast majority of crystallizations are still operated batch-wise since the process design is straightforward and can be based on the experience of many decades. Usually, a defined seeding strategy, combined with a controlled cooling policy, are utilized to meet the rigorous product specifications. This is the most common approach for process development and commercial manufacturing, but it also has various disadvantages. Batch processes suffer from batch-to-batch variations of the product and potentially high manufacturing costs, due to dead times for charging and cleaning [5–7]. To overcome these drawbacks, the crystallization can be carried out continuously, which increases the productivity and especially the space time yield. Hence, continuous operation has received increasing attention in recent years as a key element for improving crystallization based production [7–9]. Simple design methods for continuous crystallizations, like for batch-processes, are still missing. The continuous operation relies essentially on a precise control of nucleation and crystal growth. Thus, the crystal size distribution (and shape) must be well known, together with the liquid phase conditions, to reach a certain steady state, which yields the desired product.

Hence, in order to optimize existing processes, to develop new ones, or to transfer a batch-wise operation into a continuous mode, it is essential to monitor the solid phase formation together with the actual state of the liquid phase. The measurement of integral parameters, like temperature or concentration, is state of the art while determining the solid phase state, e.g., the crystal size is still quite challenging. Hence, this contribution is concerned with the evaluation of a novel inline probe for the measurement of CSD. In the first part, the technology is explained and compared to commonly applied analysis methods. In the following, the experimental approach is introduced, which is used to investigate the quality and application range of the new probe. Altogether, the results of three different techniques for the measurement of CSD are compared with the example of two substance systems in a wide range of operation conditions.

### 1.1. Particulate Measurement Techniques

The measurement of CSD is a challenging task due to the limited variety of monitoring techniques. However, the choice of the right measurement principle, with sufficient temporal and spatial resolution, can be decisive for process control and performance. With respect to this, a comprehensive summary of available techniques is given in the following paragraphs.

An overview for particulate measurement systems, with multiphase flow, in general is given elsewhere [10,11], and focusses on the determination of particulate properties like particle size and shape, flow field visualization, or concentration measurements. In contrast to the broad range of applications discussed in the cited literature, this article focusses on crystallization. One can distinguish between offline/atline, online, and inline measurement techniques. Offline measurement systems are used to analyze samples of the process in a laboratory, resulting in a relatively long deadtime between sampling and analysis results. Atline measurements are basically offline methods, that are placed close to the sampling point to minimize transportation distances. Hence, the time of the whole analytical processes is reduced. Nevertheless, both kinds of measurement operation mode are usually difficult to automate, often need manual adjustment, and are too slow for an efficient process control. For fast analysis and direct process control, online and inline measurements are indispensable.

Online measurements are commonly used together with a bypass. A representative sample is continuously withdrawn from the process into the measurement system and then returned to the

apparatus. Inline techniques, often designed as probes, collect the information at the point of interest. The acquired data can be exploited in both cases, either to adjust the process conditions or to serve as an input for actuators on a model-predictive control [11].

For the evaluation of the transient particulate phase during crystallization processes, offline/atline approaches are slow and tedious (sampling, washing, etc.), with the drawback of influence on the particle size distribution (PSD) during the involved procedure. The only alternative is inline or online determination, to directly control the product properties, which only can lead to high quality products [3,7,12,13].

The most commonly used quantitative particle measurement methods applied in crystallization are either laser diffractometer (LD) or focused beam reflectance measurement (FBRM). The LD is a classical offline technique and measures the refraction of light to determine the diameter of an equivalent sphere. Therefore, this technique is mostly used for spherical and compact particles, because it is well known that this technique struggles with non-spherical crystals, especially if these have high aspect ratios [14,15]. FBRM is the state of the art technique in crystallization, because it is commercially available and easy to use [4]. A probe that is inserted in the apparatus and measures the chord lengths of the crystals from backscattered light originating from a fast-rotating laser beam [7,16–24], yielding a one-dimensional chord length distribution (CLD) instead of a real crystal PSD. The conversion of a CLD to a PSD is based on models and assumptions, but is lacking for complex and elongated particle shapes [25–28]. Another limitation of the technique is the one-dimensional distribution, which proves insufficient at higher dimensions, like with the width and length of a needle-shaped crystal [3,22,29,30].

So far, only optical imaging measurement techniques, together with sophisticated image processing algorithms, manage to determine efficiently the particle shape of a particulate phase [13,30]. Other noninvasive niche methods rely, for instance, on the use of supersonic wave probes for the characterization of the dispersed phase [31–33].

In the simplest optical image processing approach, a camera is placed in front of a transparent reactor [34–36], e.g., a stirred glass vessel, for continuous monitoring. This concept often suffers from image distortion due to the curved reactor wall, poor contrast ratios, and limitations with respect to the solid content or suspension density. In addition, the acquired images are mostly evaluated manually, because their image quality varies in illumination and contrast. In order to benefit from online and inline techniques, it is important to have an automated, or at least fast, image processing algorithm. Therefore, it is of major importance to utilize a camera setup that acquires images with high contrast, sharp edges, and constant image quality.

Online approaches via bypass variants are costly at industrial scale and therefore commonly applied at laboratory scale. In this case, the suspension is isokinetically withdrawn from the process, passing a cuvette or flow-through cell for analysis. Hence, high contrast images with constantly sharp particles can be acquired if the focus is in the middle of the cell. Various publications have demonstrated the use of this technique for the determination of the PSD for resilient crystals that do not tend to break [37–39]. Another variant is with a stereoscopic imaging system for reconstruction of the 3D-shape of crystals [29,30,40].

Other variants are image-giving inline probes, which have been developed and established in the last twenty years [12,41–45]. These common incident-light probes are inserted into the reactor, acquiring the information at the point of interest. Available systems use entocentric lenses and are therefore quite compact, but unfortunately suffer from a small focal plane. As a result, most of the particles in the measurement area may appear blurry, which can lead to inaccurately imaged particles and an erroneous PSD. Thus, these are mostly used for qualitative analyses, like monitoring secondary nucleation or phase transformation, in combination with an FBRM that measures the quantitative particle chord length [17,18,20,23,46,47].

The ultimate approach is with tomographic methods that have excellent temporal and spatial resolution, and where applicability is possible even with high solid content and in opaque media. This measurement principle was developed in human medicine and is completely

noninvasive. Today, this technique is also applied in various fields of process engineering technology. Its disadvantages are a high space requirement and high costs, thus only at laboratory scale can applications be found. The techniques used in this article, beside sieving, are image-based techniques: an established bypass online microscope and the telecentric shadowgraphic probe for the evaluation of this new measurement technique. Both imaging techniques are described in detail below.

*1.2. Shadowgraphic Optical Probe and Online Microscope*

The in situ data of the experimental investigations in this study were recorded with two measurement systems: a commercial QICPIC online microscope (Sympatec, Germany) installed in a bypass and a shadowgraphic inline probe. The online microscope was using the transmitted light technique, where a pulsed light source is vis-à-vis of a camera. The measurement volume was formed by a flow-through cuvette that was placed between the optics and the light source. The parts were adjusted, so that the focus plane was in the middle of the cuvette. Thus, particles that were transported through the cell were captured sharply, with constant image quality.

A peristaltic pump fed the suspension via a temperature-controlled bypass to the cuvette and back to the crystallizer. The measurement volume had a fixed width of 2 mm given by the cuvette geometry. The camera and lens provided a square field of view of 5 mm in height and width at a resolution of 1024 pixel × 1024 pixel. The microscope software supports an autofocus function to adjust the focus plane of the camera in the middle of the flow cell, alternatively it could be adjusted manually.

The shadowgraphic probe is a further development of the so-called optical multimode online probe (OMOP) [48,49]. The primary design of the probe is based on a transmitted light technique and consists of two opposite protection tubes in a measurement flange.

One tube contains the illumination unit and the other one the camera and a telecentric lens. The light source consists of a LED placed in the focus point of a plano-convex lens, which emits a parallel light beam. The light beam is passed through the measurement volume between the two tubes, which are sealed with inspection windows in the front. Through the parallel light, high contrast images of the particles within the measurement volume can be acquired, even if the particles are nearly transparent.

In contrast, commercial image-based probes apply an incident light technique with an endocentric optics. In comparison, these probes have lower contrast ratios and natural image distortion is easily caused by the illumination and the optics [50,51]. Therefore, image analysis is quite challenging and requires significant effort to achieve quantitative results [52]. Furthermore, the depicted particles captured by the camera appear smaller, the larger the distance from the entocentric lens, and need sophisticated correction. Due to a comparably small focus areas, only a limited number of particles can be evaluated.

Alternatively, telecentric lenses provide a distant independent image of the particles when using parallel light. An aperture in the image-sided focus point of the lens filters out non-parallel light beams and, thus, shadowgraphic pictures are generated by parallel light only. Hence, no prior calibration is necessary. In addition, these lenses have higher depth of field compared to entocentric lenses, therefore having a large measurement volume instead of a focus plane. Hence, various versions of the OMOP for different applications, like capturing droplets [53,54], bubbles [55–57], or sprays [58,59] were developed.

In order to achieve easier access to an apparatus and to promote industrial applications, two probes have been designed recently; first, a robust DN 80 variant for industrial applications, with the full functionality of the two-sided OMOP principle [60]. Second, as a further development, a single sided endoscopic probe in DN 50 version was designed for laboratory scale [56,61], and was used for the experiments of this article. The probe has an adjustment mechanism where the position inside the apparatus and width of the measurement volume can be changed, even during the crystallization process, to adapt it to the increasing particle concentration (see Figure 1).

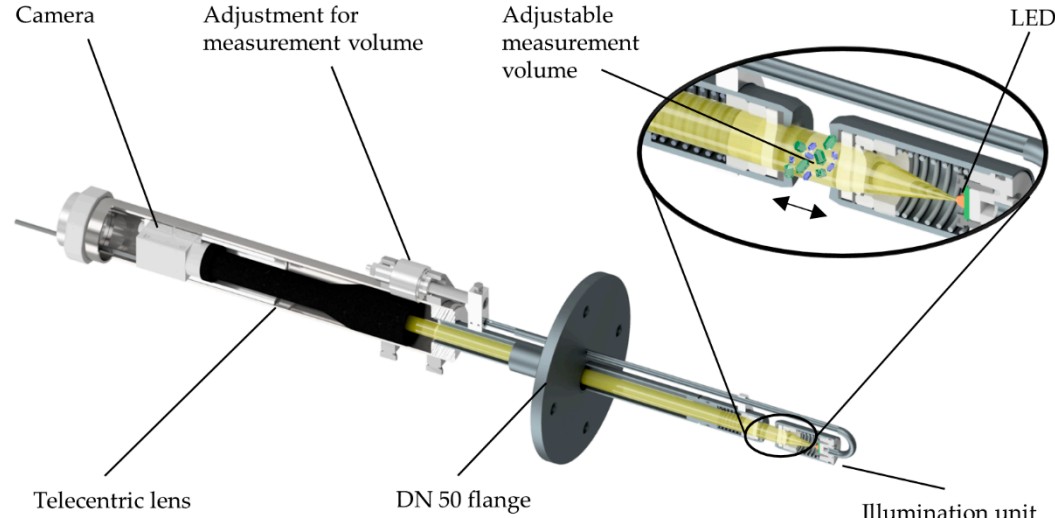

**Figure 1.** One-sided DN 50 telecentric shadowgraphic probe for laboratory scale published in [56,61].

## 2. Materials and Methods

### 2.1. Experimental Setup

For the experiments in this study, the following setup was used for all experiments (Figure 2):

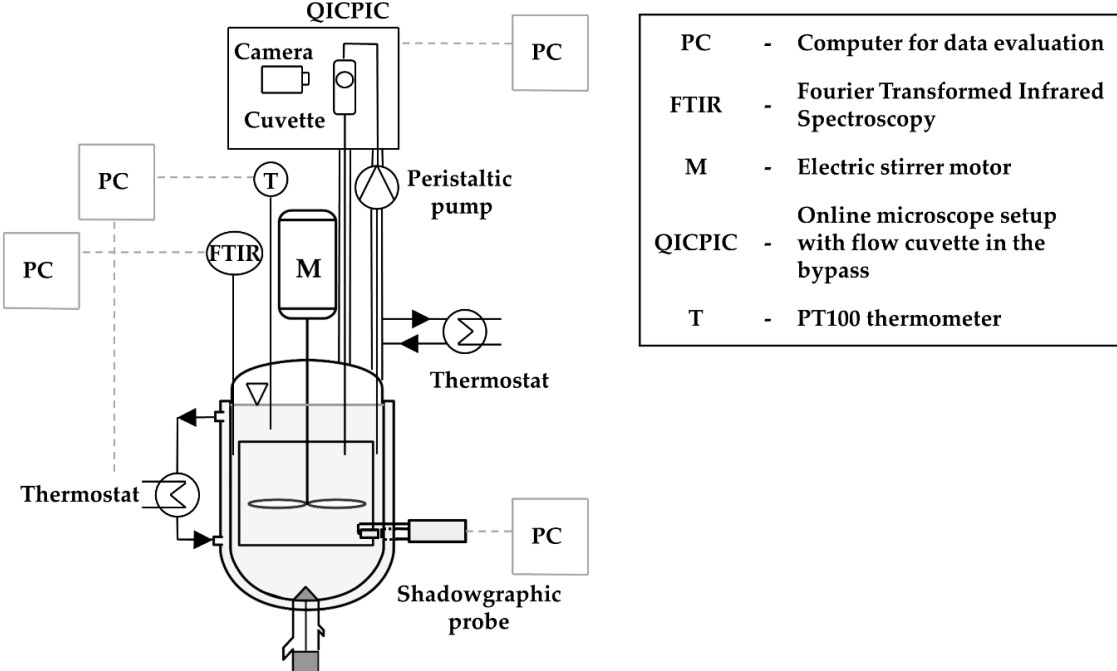

**Figure 2.** Scheme of the 25 L utilized double jacket draft tube crystallizer setup [62].

The temperature-controlled 25 L double jacket draft tube crystallizer was equipped with a propeller-type stirrer (diameter 150 mm, BASF, Ludwigshafen, Germany) and a PT100 was used to monitor the temperature (T in Figure 2). In order to measure the liquid phase composition, an attenuated total reflection Fourier-transform infrared spectroscope (ATR-FTIR, Thermo Fisher Scientific, Waltham, Massachusetts, USA) was used (FTIR in Figure 2).

The measurement depth of the shadowgraphic probe was set to 2 mm to be consistent with the flow cuvette of the online microscope. The probe was fitted with a Basler 1440-73 gm camera and a

1× telecentric lens, and the measurement volume was in the middle between the draft tube and the reactor wall. The bypass for the online microscope was inserted at the top of the reactor, directly above the shadowgraphic probe to ensure that the withdrawn suspension was similar. A high bypass flow rate (≈40 L/h) was chosen to ensure unclassified sampling of the suspension. All bypass tubes were double jacketed and temperature-controlled by a thermostat, which were set to 1–1.5 K above the reactor temperature. Details of the QICPIC bypass setup are reported elsewhere [37,62].

Both measurement techniques were capturing 750 images at 25 fps for each measurement point to monitor the executed experiments. Additionally, solid-free offline samples of the liquid phase were taken every 5 min, which were analyzed gravimetrically to verify the inline data, and finally a representative suspension sample was taken from the bottom valve at the end of each experiment to allow for sieve analyses.

## 2.2. Substances

In order to analyze different crystal shapes, potassium dihydrogen phosphate ($KH_2PO_4$) and thiamine hydrochloride were used.

$KH_2PO_4$ grew bipyramidal, prismatic shaped crystals, depicted in Figure 3a, and thiamine hydrochloride monohydrate grew needle-like shape, as depicted in Figure 3b.

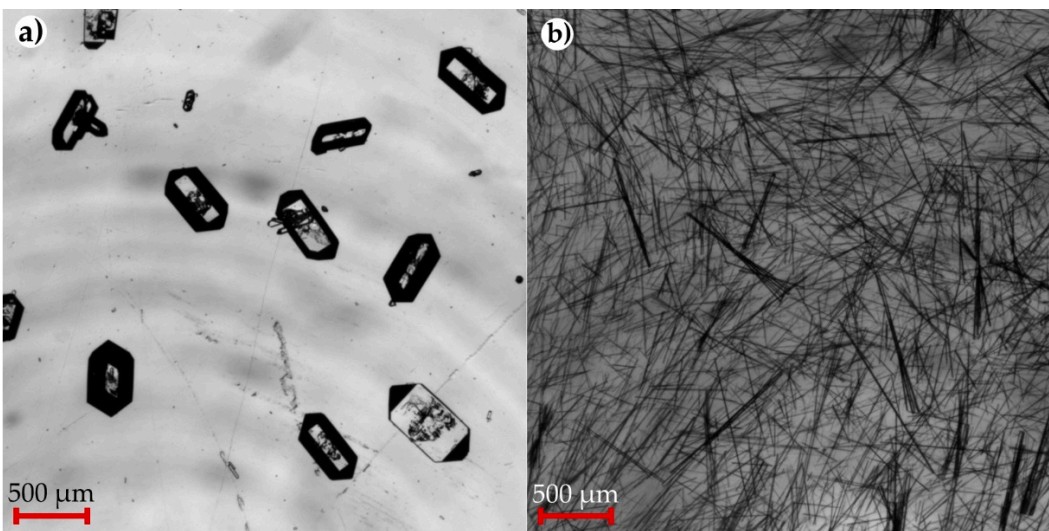

**Figure 3.** (**a**) $KH_2PO_4$ crystals; (**b**) Thiamine hydrochloride monohydrate crystals captured by the shadowgraphic probe during the experiments.

$KH_2PO_4$ grows resilient crystals, that are very suitable for validation purposes, because they do not tend to break in a bypass, during filtration, or sieving. In order to design the crystallization process, the solubility and the kinetics must be known, especially because $KH_2PO_4$ tends to form longer or shorter body prisms, depending on the operation conditions. The crystallization kinetics in aqueous solutions and the solubility are well known and reported [62,63]. The latter can be described by the following polynomial expression:

$$
\begin{aligned}
C_{sat}(T)[\text{wt.} - \%] &= 15.24 \, \text{wt.} - \% + 2.06 \times 10^{-1} \tfrac{wt.-\%}{°C} T + 1.01 \\
&\quad \times 10^{-2} \tfrac{wt.-\%}{°C^2} T^2 - 1.45 \times 10^{-4} \tfrac{wt.-\%}{°C^3} T^3 + 1.23 \\
&\quad \times 10^{-6} \tfrac{wt.-\%}{°C^4} \, T^4
\end{aligned}
\tag{1}
$$

Thiamine hydrochloride exists in five solid-state forms, according to the literature [64,65]. The desired form in industrial applications is a pseudo-monohydrate that crystallizes in contact with water, and contains 0.5 to 1 mole water per mole thiamine in the needle-like shaped crystals (see Figure 3b).

However, this monohydrate is metastable at ambient conditions and converts fast into the thermodynamic stable thiamin hydrochloride hemihydrate. Hence, the solid-liquid equilibrium of the pseudo-monohydrate is difficult to measure, and therefore barely investigated [65,66]. Fortunately, data of the less-soluble thiamine hydrochloride hemihydrate in some binary solvents are reported, and therefore the experimental design was based on data of a binary water/ethanol mixture [67]. Some physiochemical properties of the thiamin hydrochloride and $KH_2PO_4$ are shown in Table 1.

**Table 1.** Properties of the utilized substances $KH_2PO_4$ and thiamin hydrochloride.

| Substance Properties | Symbol | $KH_2PO_4$ | Thiamine HCl Anhydrate | Unit |
|:---:|:---:|:---:|:---:|:---:|
| **Solid density** | $\rho_{solid}$ | 2340 | 1.4 | [kg/m$^3$] |
| **Molar mass** | M | 136.09 | 337.27 | [g/mol] |
| **Purity** | Pr | ≥99 | ≥98 | [%] |
| **Vendor** | | Applichem | Sigma Aldrich | |

### 2.3. Experimental Procedure

Table 2 depicts the conditions of the executed experiments, one with thiamine hydrochloride (Exp. 5) and four with $KH_2PO_4$ (Exp. 1–4). For Exp. 1–3 the saturation temperature was 35 °C and the cooling rate, the final temperature, and the mass of the seed loading was varied for each experiment to evaluate the limits, with respect to the suspension density and crystal size. The seed fraction was sieved and had a normal distributed initial size of 212–300 µm for all experiments, except for Exp. 3. Smaller seeds with a range of 150–212 µm were used in this case to alter the initial suspension and optical density, to evaluate the impact of these parameters on the crystal size measurements. The fourth experiment was carried out with a larger initial concentration (according to a saturation temperature of 56.5 °C), since it is well known that the bypass of online microscopes often tends to block under these conditions. In addition, an anti-solvent crystallization, Exp. 5, of thiamine hydrochloride was performed to evaluate the applicability of the shadowgraphic probe to fragile crystal systems.

**Table 2.** Process conditions of the performed experiments with $KH_2PO_4$ and thiamine hydrochloride.

| Exp. | $\Delta T/\Delta t$ [°C/h] | $T_{Sat}$ [°C] | $T_{Seed}$ [°C] | $T_{end}$ [°C] | $m_{H2O}$ [°C] | $m_{EtOH}$ [kg] | $m_{solute}$ [kg] | $m_{seeds}$ [kg] |
|:---:|:---:|:---:|:---:|:---:|:---:|:---:|:---:|:---:|
| **Exp. 1—$KH_2PO_4$** | −7.5 | 35 | 34 | 27 | 21 | - | 6.5 | 0.05 |
| **Exp. 2—$KH_2PO_4$** | −10 | 35 | 34 | 15.5 | 21 | - | 6.5 | 0.1 |
| **Exp. 3—$KH_2PO_4$** | −10 | 35 | 34 | 22 | 21 | - | 6.5 | 0.2 |
| **Exp. 4—$KH_2PO_4$** | −12 | 56.5 | 56.4 | 43 | 21 | - | 9.5 | 0.1 |
| **Exp. 5—thiamine hydrochl.** | - | ≈30 | - | 25 | 5.6 | 12 | 4 | - |

$KH_2PO_4$ was added to the reactor according to Equation (1) (see Table 2), the impeller speed was set to 250 rpm and the crystallizer was heated a priori for 0.5 h to 5–10 K above the respective saturation temperature to ensure complete dissolution and equal starting conditions for all experiments. Afterwards, the clear solutions were slightly subcooled (0.1–1 K) and the seeds were added at temperature $T_{seed}$ (see Table 2) and time t = 0 h. Subsequently, cooling was executed as a simple linear cooling ramp after the seed's addition, with a certain slope between −7.5 °C/h and −12 °C/h (see Figure 4a for an exemplary temperature curve of Exp. 3—$KH_2PO_4$). Immediately after seeding (t = 0 h), the particle size was measured simultaneously every 5 min by the online microscope and the shadowgraphic probe. The experiments ended at the final temperature, $T_{end}$, if either the suspension density was too high, resulting in too much overlapping of the single crystals, or excessive nucleation was observed. During the last measurement of the crystal size by the optical methods an unclassified suspension sample was taken from the bottom valve of the reactor, and was immediately filtered using a strainer and a filter paper. Then, the filter cake was washed with an adjusted ethanol/water-mixture to prevent nucleation or dissolution of the crystals through the residual

mother liquor. Afterwards, the crystals were dried and sieved to determine the mass-based size distribution. During the experiments with $KH_2PO_4$, the state of the liquid phase was monitored by a calibrated ATR-FTIR and by solid-free liquid samples, which were taken every 5 min, simultaneous to the particle size measurements (see Figure 4b).

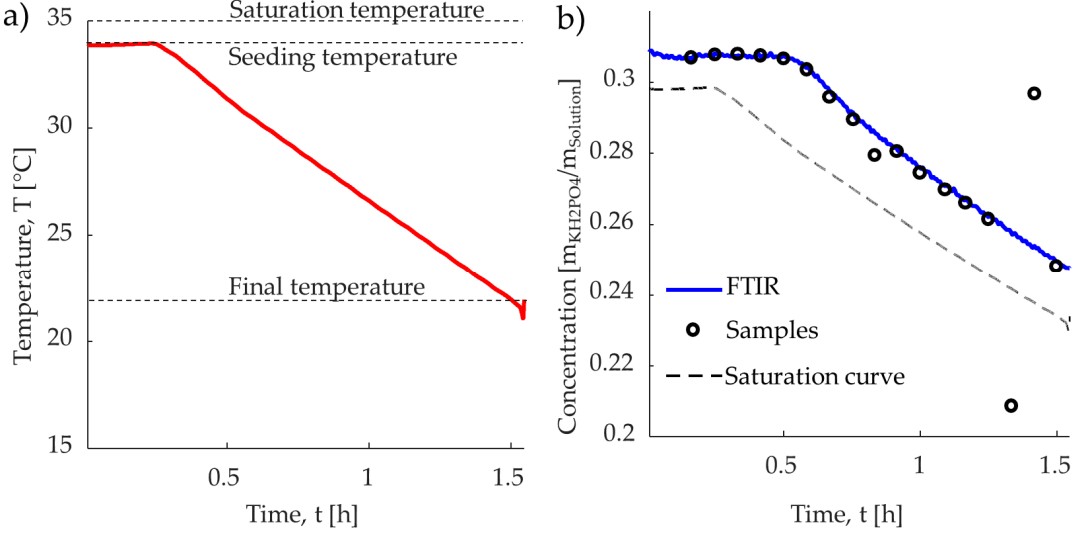

**Figure 4.** (**a**) Temperature profile of Exp. 3—$KH_2PO_4$: saturation, seeding, linear cooling ramp, and final temperature. (**b**) Concentration profile of Exp. 3—$KH_2PO_4$: FTIR, offline samples, and saturation curve.

From ATR-FTIR spectra the mass fraction ($m_{KDP}/m_{solution}$) was calculated by an existing calibration, successfully applied in the past [68] to evaluate the suspension density, where $c_{FTIR}$ is the concentration of the FTIR and $c_0$ is the initial concentration:

$$\rho_{Susp}(t) = (c_0 - c_{FTIR}(t)) + \frac{m_{seed}}{m_{solution}} \tag{2}$$

$$S(t) = \frac{c_{FTIR}(t)}{c_{sat}(T(t))} \tag{3}$$

The supersaturation, $S$, is the driving force in crystallization, and was calculated according to Equation (3), with the concentration at saturation, $c_{sat}(T(t))$.

The experiments started at small supersaturations, which gradually increased due to cooling. After about 0.6 h sufficient solid surface was present in the crystallizer to counterbalance the supersaturation generation and the driving force started to decrease, exemplarily shown in Figure 5a for Exp. 3—$KH_2PO_4$.

The concentration of the liquid phase versus temperature in the binary phase diagram for all $KH_2PO_4$ experiments is given in Figure 5b. Exp.1–3 have almost identical conditions with a saturation temperature of 35 °C and only Exp. 4 was saturated at an elevated temperature. A significant influence of the seed load or the cooling ramp on the concentration profile is not clearly visible due to the fast crystallization kinetics of $KH_2PO_4$.

In addition, a fifth experiment with thiamin hydrochloride was carried out as an anti-solvent crystallization via primary nucleation. The thiamine hydrochloride was dissolved in water and added to the reactor. At $t_{start} = 0$ h the ethanol was added as the antisolvent. The initial masses were calculated based on literature data [67]. The experiment was carried out at a constant temperature of 25 °C, and the impeller speed was set to 250 rpm, similar to the experiments with $KH_2PO_4$.

The shadowgraphic probe took pictures of the suspension every minute from the beginning, and after a significant number of crystals were visible (t = 1 h), the online microscope in the bypass was put into operation.

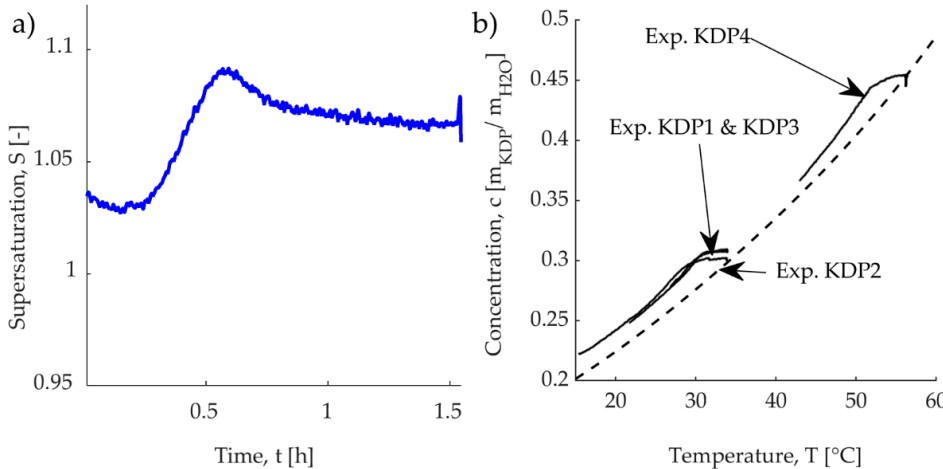

**Figure 5.** (**a**) Supersaturation profile calculated for Exp. 3—$KH_2PO_4$ with the FTIR data, (**b**) Liquid phase concentration for Exp. 1–4 $KH_2PO_4$, measured with the FTIR depicted in a part of the binary phase diagram of $KH_2PO_4/H_2O$. The dashed line is the saturation curve according to Equation (1).

The experiment was carried out until the concentration was too high to identify single crystals. Taking a suspension sample similar to the experiments with $KH_2PO_4$, was not possible since the thin, needle-like crystals of the thiamine hydrochloride monohydrate broke during filtration and further handling. A representative sieving analysis was, therefore, not possible and the crystal length and width were only determined via image processing.

### 2.4. Image Processing

The pictures of the probe and the microscope were evaluated with an existing MATLAB routine [69]. Based on the difference in contrast between the crystals and the background, the crystals could be isolated through contrast enhancement and binarization. In addition, a dynamic background subtraction out of a picture series was carried out to eliminate scratches or immobile adherent particles. The $KH_2PO_4$ crystals, especially, have large bright areas in the crystal center (see Figure 3a) that would lead to erroneous object identification. Therefore, morphological closing and region filling was utilized to fill the empty areas within the crystals.

The evaluation focuses on single crystals only, assuming agglomerates are of negligible number. Therefore, two shape descriptors, the numerical eccentricity, $\varepsilon$, (Equation (4)) of an ellipse, and solidity, s, for the description of the convexity (Equation (5)), were used to exclude agglomerates from further evaluation.

$$\varepsilon = \frac{\sqrt{a^2 + b^2}}{a} \tag{4}$$

$$s = \frac{A}{A_{convex}} \tag{5}$$

The use of these parameters was based on experience and was successfully applied for single crystals of $KH_2PO_4$ in the past. Particles are classified as single crystals if the eccentricity was within 0.4 to 1, and the solidity from 0.95 to 1, respectively. As a result, gas bubbles and overlapping crystals were excluded in the data evaluation. This was manually crosschecked by comparing the crystal detection results with the original images. Furthermore, crystals touching the image border were not considered in the evaluation, since incomplete objects lead to erroneous size calculations. More details about the algorithm are reported in the literature [69].

In order to evaluate the geometry of the observed objects and to calculate meaningful distributions, it is necessary to choose a reasonable characteristic length. For slightly elongated bipyramidal crystals with a square prism body, the width of the square cross-sectional area can be used as a characteristic length, L, to describe its size. This length can be obtained by the minimal Feret's diameter, $L_{Fmin}$ of

the projected area of the crystal, but the orientation of the crystal to the picture must be considered. The relevant orientation for the representation of the minimal Feret's diameter is given by the rotation around an axis, passing the two pyramid tips along the elongated direction of the crystal. Imaging a rotation around this axis, while constantly measuring the minimal Feret's diameter, gives values between a minimum and a maximum value, $L$ and $\sqrt{2}L$ for $L_{Fmin}$. Assuming that the orientation of the particles is normally distributed, for the probability of all rotations between the two extreme values, an arithmetic mean $L_1$ can be described according to:

$$L \approx L_1 = \frac{2\,L_{Fmin}}{1 + \sqrt{2}} \tag{6}$$

Based on this averaged crystal width of the square prism body, the distributions can be compared with the sieving analysis, where $L_1$ is measured. There are advanced methods for the correction of the crystal orientation reported in the literature [36], but for the sake of simplicity and computational effort, this approach was chosen.

The single crystals detected and measured by the image analysis were sorted in 400 size classes between 1 and 2000 μm, based on their corrected crystal size, and normalized by the class width, giving the number distribution, $f(L)$. Normalization by the integral $\int f_j(L)dL$ yields the density distribution $q_j$ of dimension $j$:

$$q_j = \frac{fj(L)}{\int f_j(L)dL} \tag{7}$$

Additionally, the percentiles, $p$, were utilized for the comparison of the online microscope and the shadowgraphic probe's results according to:

$$Q_j\big(L_p\big) = \int_0^{Lp} q_j(L)dL = p \tag{8}$$

The results were depicted and evaluated in terms of number distribution ($j = 0$) and mass distribution ($j = 3$), mainly, but other characteristic values of distributions can be used, as well [70,71].

Thiamin hydrochloride crystals were characterized utilizing the same algorithm, but without any correction of the Feret's diameter for the orientation. The minimal and maximal Feret's diameter were interpreted as the width and the length of the needles.

Further, for both substance systems an optical density was calculated based on the acquired and binarized images of the shadowgraphic probe and the QICPIC. For this purpose, the number of all black pixels in an image was divided by its resolution. This ratio was used in the following, called optical suspension density, and helps to interpret the results.

## 3. Results and Discussion

In the following, the results of Exp. 3 with the highest initial suspension density will be discussed in detail, because the Exp. 1 and Exp. 2 show similar results and have the same starting saturation temperature. Furthermore, Exp. 4 will be shown, as it has a higher starting saturation temperature. Afterwards the results of the thiamine hydrochloride crystals will be discussed.

### 3.1. Comparison of the Crystal Size Measurement Techniques with $KH_2PO_4$ Crystals

Figure 6 gives an example of captured $KH_2PO_4$ crystals, with the online microscope in comparison with the shadowgraphic probe. Both pictures show good contrast ratios, which simplifies the subsequent image processing. The imaged crystals have clear edges and can be accurately detected and measured by the algorithm. Thus, the number, $q_0$, and mass, $q_3$, density functions of the distributions could be calculated as shown in Figure 7. In the number density functions (see Figure 7a,b), the evolution of the fines content can be visualized, while the mass density function (see Figure 7c,d) serves to illustrate the evolution of the larger crystal fractions.

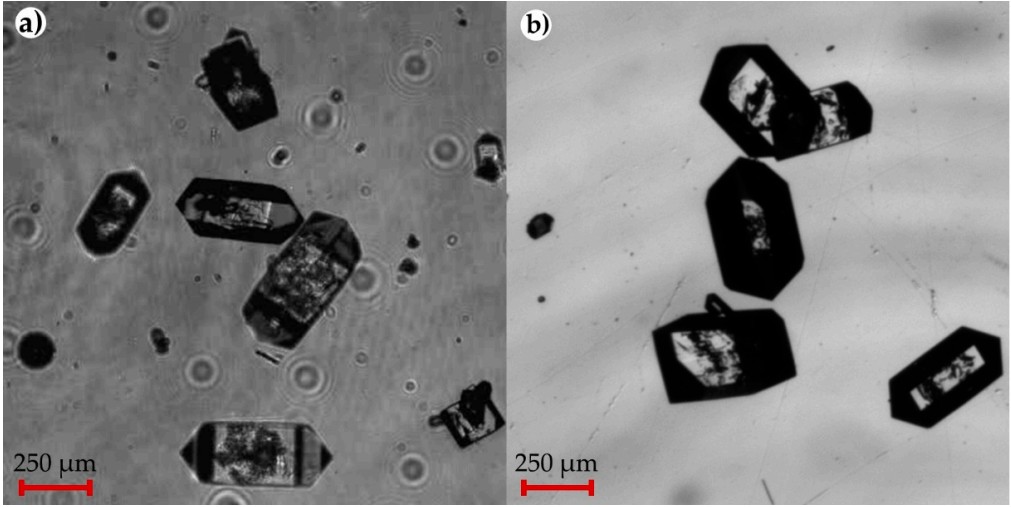

**Figure 6.** KH$_2$PO$_4$ crystals captured with (**a**) QICPIC (online microscope with bypass); (**b**) shadowgraphic probe (OMOP, inline probe). The images are enlarged for a better view of the edges.

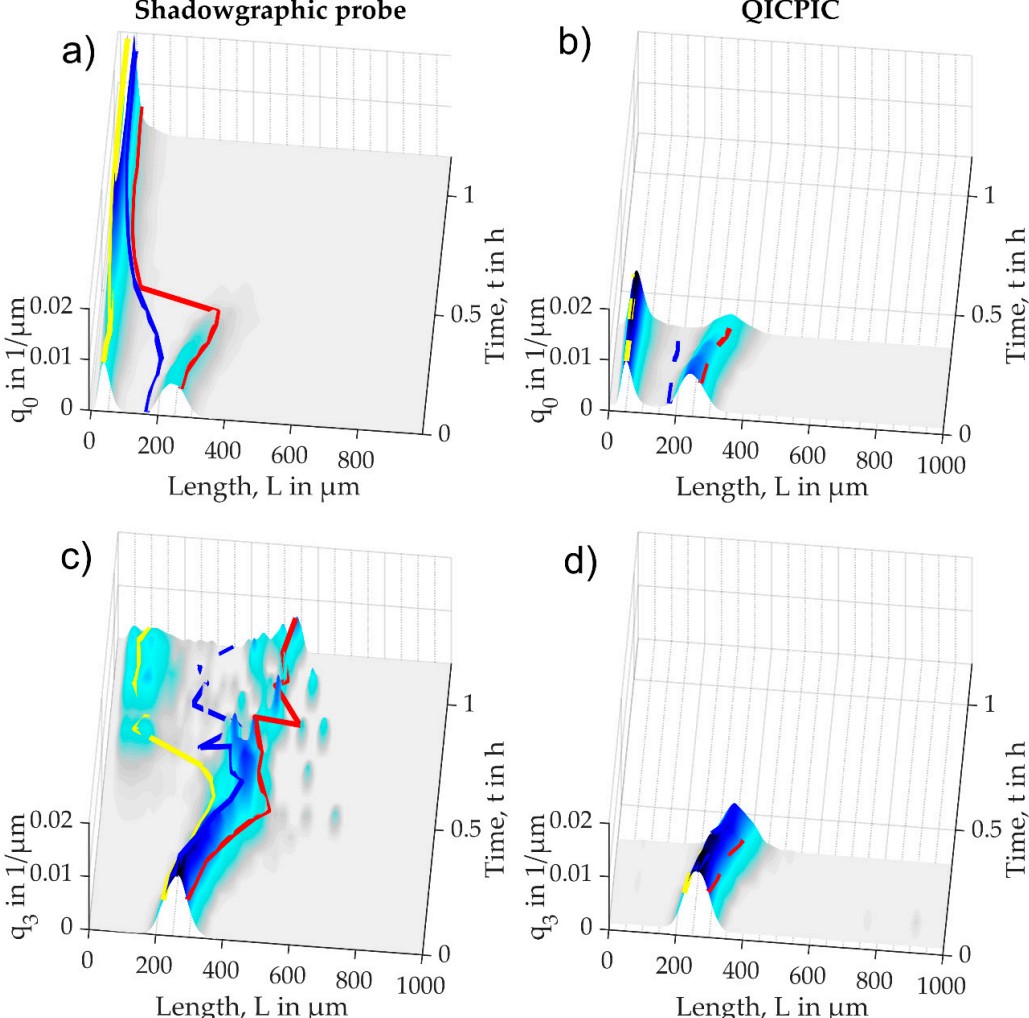

**Figure 7.** KH$_2$PO$_4$—Exp. 3 distributions (**a**) $q_0$-distribution shadowgraphic probe; (**b**) $q_0$-distribution QICPIC; (**c**) $q_3$-distribution shadowgraphic probe; (**d**) $q_3$-distribution QICPIC. *Solid lines*—percentiles of the shadowgraphic probe; *Dashed lines*—percentiles of the QICPIC; *yellow and red* −0.15 and 0.85 percentile distribution; *blue*—transient mean sizes of the distribution.

In general, both techniques give similar distributions. The number distributions show, for t = 0 h two major crystal fractions, one with about 50 μm and another one with 184 μm. As seen in the mass distributions, the small fraction is not visible and was probably caused by fine grain $KH_2PO_4$ particles in the seeds and dust. After a time of about t = 0.6 h, where the largest supersaturation was present (see. Figure 5a) a shift in crystal size towards bigger crystals due to growth can be observed (see Figure 7). Obviously, a certain threshold driving force must be present for the seeds to become active. Several reasons are known for this behavior, e.g., the crystal surfaces need to heal before macroscopic growth can take place or impurities block growth centers. However, a detailed study of the mechanism is not the focus of this article. The $q_0$-distributions (Figure 7a,b) also show that the number of smaller particles increases at the same time, caused by nucleation. After crystal growth can be observed, a significant broadening of the seed fraction is visible (see Figure 7c,d for t > 0.6 h), which can be attributed to growth rate dispersion. This influence can also be seen with a look at the percentiles, therefore they are shown as the top view in Figure 8, on the 3D diagram in Figure 7. For a better overview the surface plot is not shown in Figure 8.

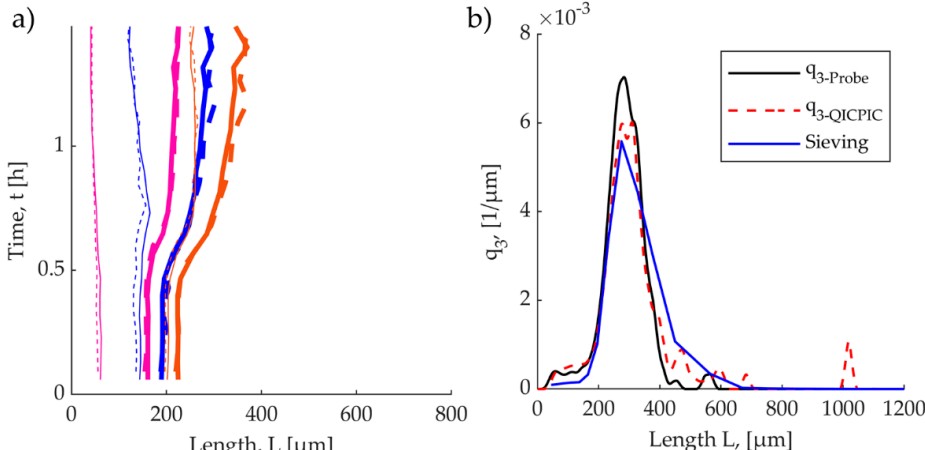

**Figure 8.** (**a**) Percentiles and mean values of the crystal size distributions from Figure 7 *thin lines*—percentiles of the $q_0$-distribution; *bold lines*—percentiles of the $q_3$-distributions; *Solid lines*—percentiles and mean values shadowgraphic probe; *dashed lines*—percentiles and mean values QICPIC; *brown; magenta* −0.15 and 0.85 percentile of the corresponding distribution; *blue*—mean size of the corresponding distribution. (**b**) Comparison of the last mass distribution: shadowgraphic probe, QICPIC, sieve analysis.

The percentiles of the number, and mass density distribution, match well for both optical measurement techniques (Figure 8a), and show almost identical curves. Hence, an explicit classification effect of one measurement technique, either caused by the sample withdrawal to the bypass or by the measurement gap of the inline probe, can be excluded. For t = 0 h, the percentiles match with the initial size range of the seeds (see Table 3), and confirm a reasonable measurement of the crystal size. The change of the crystal size can be tracked properly over the whole experimental time. This can be confirmed with respect to the $q_3$-distributions obtained by the sieve analysis of the suspension sample at the end of the experiment (see Figure 8b).

**Table 3.** Mass-averaged crystal widths evaluated with the optical measurement techniques at the start and the end on Exp. 1–4 $KH_2PO_4$.

| Exp. no. | Seed Size [μm] | Experimental Time [h] | Mass-Averaged Crystal Width *L* [μm] at the Start | | Mass-Averaged Crystal Width *L* [μm] at the End | | |
| --- | --- | --- | --- | --- | --- | --- | --- |
| | | | QICPIC | Probe | QICPIC | Probe | Sieving |
| Exp. 1 | 212–300 | 1.15 | 257 | 280 | 590 | 584 | 596 |
| Exp. 2 | 212–300 | 2.01 | 248 | 257 | 645 | 582 | 660 |
| Exp. 3 | 150–212 | 1.42 | 189 | 188 | 303 | 284 | 281 |
| Exp. 4 | 212–300 | 1.18 | 258 | 259 | - | 347 | 526 |

The comparison shows that the distributions measured by the optical measurement techniques fit well with the sieve analyses. The fraction of 0–200 μm is underestimated by the sieve analysis compared to the optical techniques. Probably, a part of the fines is lost during solid/liquid separation, washing, and sieving. In the range of 450–600 μm a slightly higher density for larger particles in the sieve analysis is visible. Since the image analysis focuses on single crystals, the agglomerates are not considered in the imaging techniques. In contrast, the sieve analysis also has agglomerates in the distribution, therefore this shift can be addressed to a small amount of agglomerates present in the sample. In summary, both optical measurement techniques are suitable to evaluate crystal size distributions. The deviations are in the typical error range, except for the final crystal size of Exp. 2. However, the main sources of deviation in image-based size determination, in general, is the image conversion and the binarization. About two pixels on the edges was the common deviation during the capturing by the camera, and an additional two pixel uncertainty occurred during the thresholding for the binarization. This sums up to four pixel in total, which equals 20 μm with a pixel size of 5 μm (depending on the camera and the lens used) for both techniques, and is the typical error range for image-based size evaluation in general.

Several parameters were changed during experiments Exp. 1–3, initial seed loading, seed size, final process temperature, and the final crystal size, as well as the optical and suspension density. The latter two cannot be controlled directly but are a result of various process parameters. None of the changes led to a significant impact on the deviation between both optical measurement techniques, since a good agreement was found for density functions of all experiments (see Appendix A for the other detailed results of Exp. 1 and 2). A comparison of the initial seed sizes of all experiments (see Table 3) confirms a suitable determination of the crystal sizes between the QICPIC and the shadowgraphic probe. At the end of the experiments where larger crystals occurred, the probe measured slightly smaller crystal sizes than the QICPIC and the sieve analysis, but the deviations were still in the deviation of 20 μm mentioned above. Nevertheless, an effect of the measurement window of the shadowgraphic probe can be assumed. Larger particles tend to touch the image border, especially if the measurement window is smaller. Because the QICPIC has a larger measurement window (5 mm × 5 mm) than the probe (5 mm × 3.5 mm), this effect is maybe noticeable. Only Exp. 2 shows significant deviations for the measured final crystal sizes. For this experiment the percentiles (see Appendix A Figure A7) are almost identical for both techniques, except at the last two measurement points. For these measurements the percentiles show a significant drop, and the amount of measured crystals increases drastically. The supersaturation curve (see Appendix A Figure A8) shows an increase of the concentration and secondary nucleation occurs, which causes the decrease in the mean crystal size. Additionally, the probe has more agglomerates in the images, leading to fewer single particles being detected. A classifying effect may occur within the QICPIC bypass, where these agglomerates are seen less often, and therefore more single crystals are detected. Although the final crystal sizes show deviations in Exp. 2, the percentiles confirm a suitable transient crystal size determination up to the last 10 min. It is not clear if secondary nucleation will affect the measurement in general and this must be clarified in further investigations. The usage of larger measurement windows with a sophisticated algorithm for agglomerates can maybe solve this issue.

It is important to consider the number of particles measured in order to have a statistically verified PSD. Therefore, the total amount of measured crystals for each optical technique is shown, with their corresponding optical density, in Figure 9a. For the online microscope and the shadowgraphic probe it is clearly visible that the number of measured particles decreases over time, which is caused by the increasing number of larger crystals. This is a key issue of image analysis in general as there are particles in the system that overlap with smaller particles or other particles. Hence, these overlapping clusters of particles cannot be evaluated by the algorithm, which then leads to erroneous PSD's. The other reason is, that comparable larger particles, with respect to the image size, have a higher probability of being cut off by the measurement window. Therefore, these particles are likewise not detected and lead to a smaller number of measured particles. Nevertheless, both optical techniques measure

a few thousand particles for each distribution, guaranteeing a statistically sufficient amount for a representative distribution. The results also show that the online microscope detects more particles than the shadowgraphic probe. This is an expectable phenomenon, since the measurement window of the probe is smaller, due to a smaller camera sensor size, in comparison to the microscope.

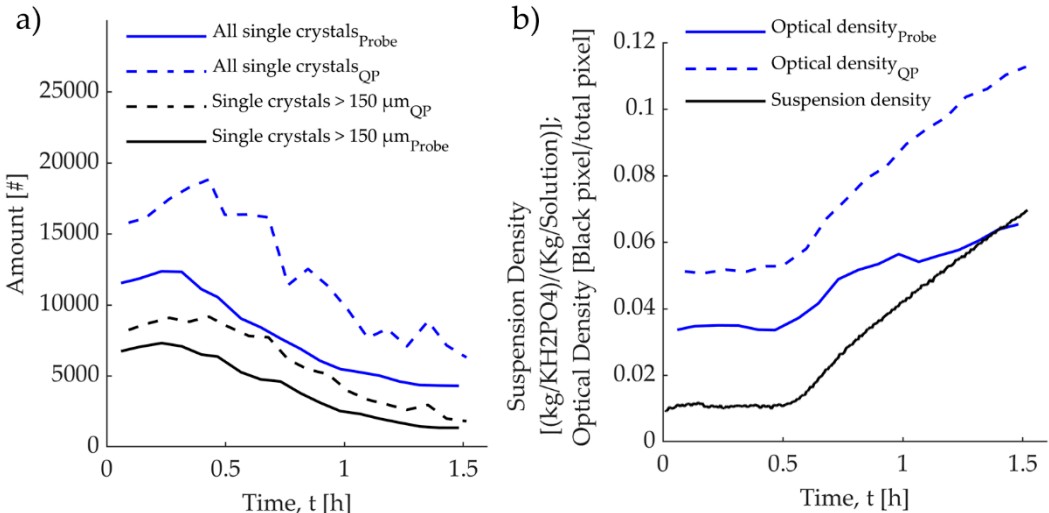

**Figure 9.** (**a**) Quantity of single crystals analyzed (**b**) the optical and suspension densities during experiment Exp. 3—$KH_2PO_4$.

The suspension density, according to Equation (2), derived from ATR-FTIR data, confirms the measured increase in crystal size at t = 0.6 h. The optical density based on pixel ratios shows a similar trend (see Figure 9b), although the optical densities do not match with the mass-based suspension density. Effects such as the overshadowing of smaller particles caused by larger ones, and overlapping, affect these values measured with the optical techniques. Therefore, the optical density is additionally connected to the dispersity of the particulate phase. Furthermore, the suspension density, determined by the concentration measurement, is a global value, while the optical measurement techniques provide local information. This means that the optical methods can recognize overall trends in the suspension density but are not suitable for its representation. Nevertheless, it could be shown that crystal size evaluation is possible and not affected by the suspension density, at least up to 6% in Exp. 3, and up to 8% in Exp. 2.

The suspension density can either be over- or underestimated with optical methods in comparison with the suspension density calculated with the FTIR data of Exp. 2, as given in Figure 10. At the start of the experiment, where a narrow distribution of one crystal size was present, the optical suspension density was less that the mass-based suspension density. This changed during the experiment, because the optical density was additionally connected to the dispersity of the system. Multimodal distributions increase the optical density, especially the fine particle content increases the particulate content in the pictures, and lead to higher optical densities. As a result, the optical density cannot be used to determine the suspension density directly, because the particulate state must be taken into account as well.

### 3.2. Investigation of $KH_2PO_4$ at Elevated Temperature

The experiment EXP. 4—$KH_2PO_4$ started at an elevated seeding temperature of 56.4 °C. After seeding, both optical measurement techniques were able to measure the initial crystal size distribution. After t = 0.5 h the reactor reached a temperature of around 50 °C and crystallization occurred in the bypass of the online microscope, which led to a blockage. Therefore, the bypass was closed down and only the shadowgraphic probe was used to evaluate the state of the particulate phase (see Figure 11).

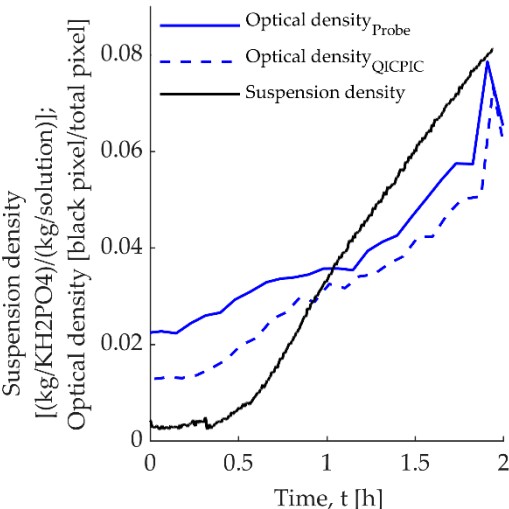

**Figure 10.** ATR-FTIR suspension density in comparison to the optical densities (microscope and probe) based on pixel ratios for Exp. 2—KH$_2$PO$_4$.

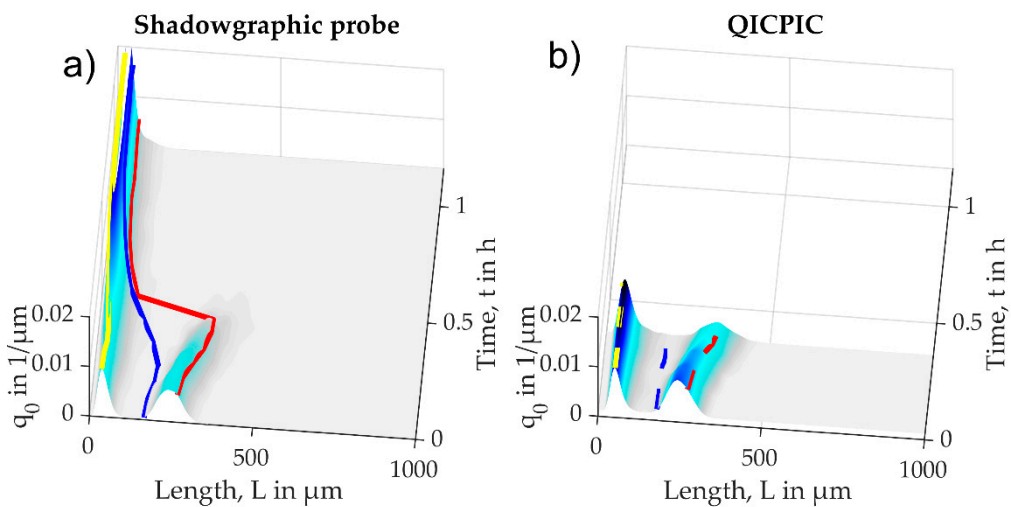

**Figure 11.** $q_0$-distributions of Exp. 4—KH$_2$PO$_4$ (**a**) shadowgraphic probe; (**b**) QICPIC; *Solid lines—percentiles shadowgraphic probe; Dashed lines—percentiles QICPIC; yellow and red −0.15 and 0.85 percentile distribution; blue—transient mean sizes of the distribution.*

Both principles have a good match with the distributions of the seeds, and similar growth is visible for both techniques at starting conditions. After the bypass was blocked, the shadowgraphic probe recognized a broadening of the mass-based distribution, therefore the larger fractions are no longer visible in the number distribution. This is also caused by nucleation of smaller crystals that are dominant in number compared to the larger grown seeds. This smaller fraction increased rapidly in number and therefore, accounts for about 15% of solid fraction in the mass-based distribution. To sum up, the inline probe can be utilized at conditions where massive nucleation and fast crystal growth leads to blocking of a bypass-based measurement technique, which requires a precise temperature control when withdrawing samples. This shows clearly that the probe opens a new field of applications, where other measurement systems fail.

### 3.3. Crystallization of Thiamin Hydrochloride Monohydrate

The crystallization of thiamine hydrochloride monohydrate was performed as nucleation from aqueous solution with ethanol as antisolvent. Nucleation was observed by the shadowgraphic probe after approximately t = 0.3 h after adding the antisolvent. These crystals were only a few pixels in

width, and a certain time was necessary to overcome the lower detection limit of the probe, therefore an exact time cannot be referred. After t = 0.6 h, a representative amount of crystals was visible within the images. The amount increased significantly up to the time of t = 1 h, when the bypass was put into operation. As the suspension passed the flow cuvette, the online microscope was not capable of setting an autofocus automatically, therefore the focus had to be adjusted manually. An example of the images taken by the optical methods is given in Figure 12.

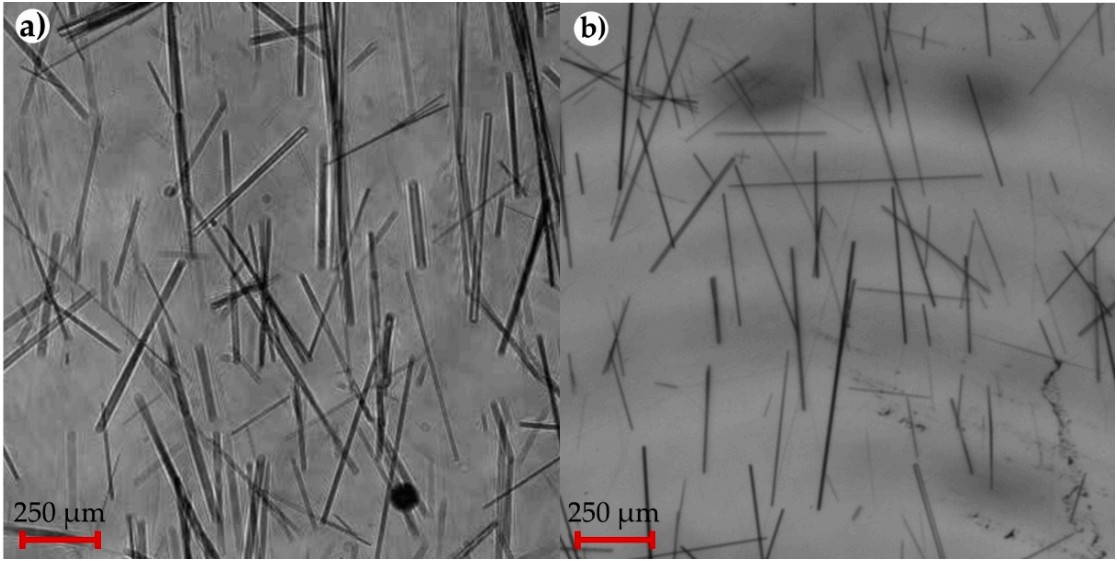

**Figure 12.** Thiamine hydrochloride (t = 1 h) monohydrate crystals (**a**) online microscope (**b**) shadowgraphic probe. The images are enlarged for a better view of the edges.

The crystals captured with the online microscope appear blurry, without clear edges, and a proper image evaluation is not possible. In addition, the bypass could only be utilized for a few minutes before the suspension flow blocked. In contrast, the shadowgraphic probe was still capable of capturing images with sharp edges and suitable image quality for a crystal size evaluation. For the present case, the telecentric lens in the shadowgrphic probe shows a clear advantage, because no focus has to be adjusted, and the depth of field covers the whole measured volume. After t = 1.22 h, the experiment was ended due to a significant increase in suspension density. At this point even a qualitative evaluation of the captured images was not possible, either with the online microscope nor with the shadowgraphic probe. Figure 13 shows images captured by the shadowgraphic probe of the suspension in different states.

At about t = 1 h the crystals grew as thin needles, as described above. They varied in length and width but were mostly isolated single crystals. Already a few minutes later (t = 1.16 h), the suspension density increased significantly, which may have been supported by natural breakage and secondary nucleation. Hence, the broken crystal pieces increase the total particle number, additionally. Due to the increased number of crystals, single particles have an increased probability of colliding with each other and forming agglomerates, which can be seen at the time mark for t = 1 h in Figure 13c,d.

For this state of the system an image evaluation of the crystals is quite challenging and may not be solved with a conventionally image analysis based on binary object identification, because the thin needles overlap and single crystals cannot be identified [72]. Interestingly, the needles tend to align with the flow direction in the measured volume of the shadowgraphic probe, especially at a higher solid content. Because the gap is comparably small to the vessel, the flow inside the gap is hindered and mostly laminar, even if the flow around it is turbulent. A group of researchers reported an algorithm which was utilized to determine the size of high-aspect-ratio crystals. They found that an irregular alignment hinders clear object detection at a higher solid content [72]. Therefore, the alignment of

the needle-like crystals in the shadowgraphic probe could be an opportunity to investigate the crystallization in such difficult systems, with high aspect ratios of the particles.

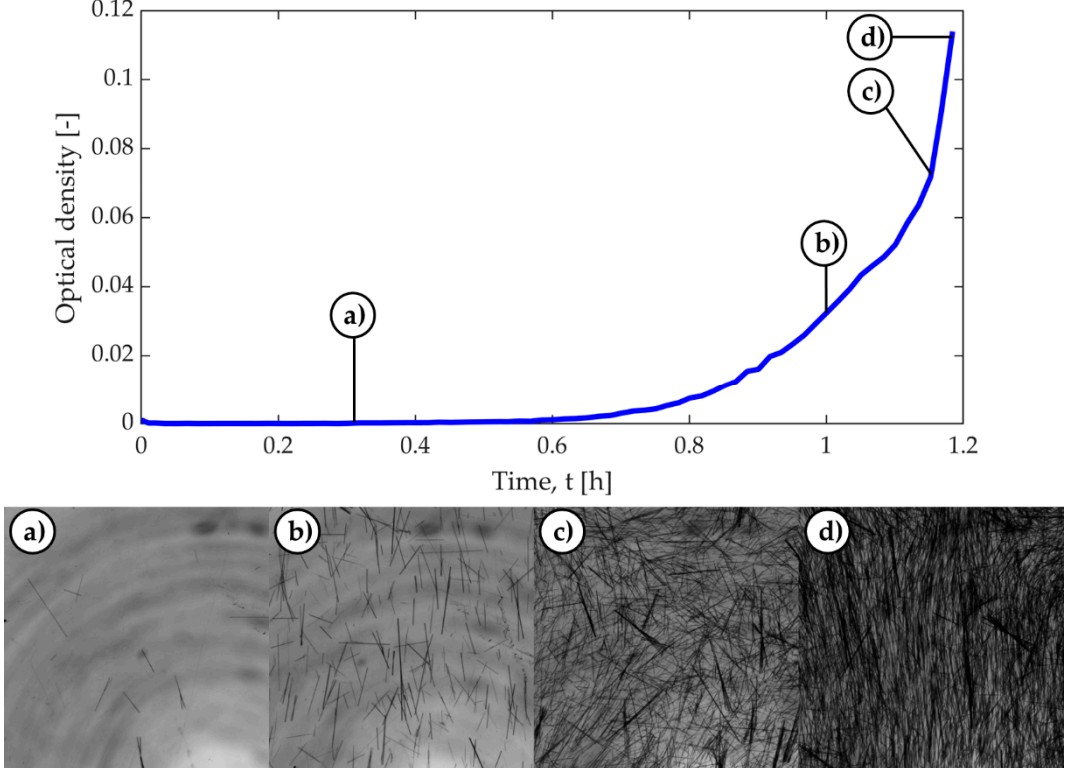

**Figure 13.** Optical density for the thiamine anti solvent crystallization based on pixel ratios of the shadowgraphic probe images. The images at the bottom were captured with the shadowgraphic probe at different experiment times; (**a**) t = 0.3 h first single crystals, (**b**) t = 1 h recognizable number of crystals, (**c**) t = 1.16 h first agglomerates and increased overlapping of the crystals and (**d**) t = 1.18 h last possible measurement point, afterwards the suspension density was too high to capture further images.

Although a crystal size determination with the presented methods was not possible after t = 1 h, a measurement of the optical suspension density was still possible (see Figure 13).

The optical density shows a rapidly increasing amount of crystals that started at around t = 0.3 h. From that point, the optical density increased with exponential progression. The width and length distributions at different experiment times are shown in Figure 14.

The diagram for t = 0 h was made as a reference, where mostly dust was detected. At t = 0.63 h the first reliable distribution shows that the single needle-like crystals have between 500–1000 μm in length and 20–40 μm width. At t = 0.97 h the crystal sizes are about the same, while their amount has significantly increased. The last diagram at t = 1.18 h depicts only a detection of small particles, which clearly shows that the simple image algorithm that was applied failed to isolate the crystals, and is the limit in PSD evaluation, at least with the methods that were used.

The comparison between the thiamine hydrochloride and $KH_2PO_4$ experiments shows clearly that the limiting optical density for image-based measurement systems depends on several properties of the particulate phase, e.g., size, size distribution, and shape. $KH_2PO_4$ could be measured up to an optical density of 8% for the shadowgraphic probe and 11% for the online microscope, with a suspension density (FTIR) of 7%. The measurement had already failed for thiamine at a value of 3% to 4% optical density of the probe (see Figure 13, t = 1 h). Hence, a clear suspension, or optical density limit, of application for both utilized techniques cannot be given here. It has to be determined for each substance system individually, and the methods for the image evaluation must be adjusted for the specific case in order to maximize the applicability.

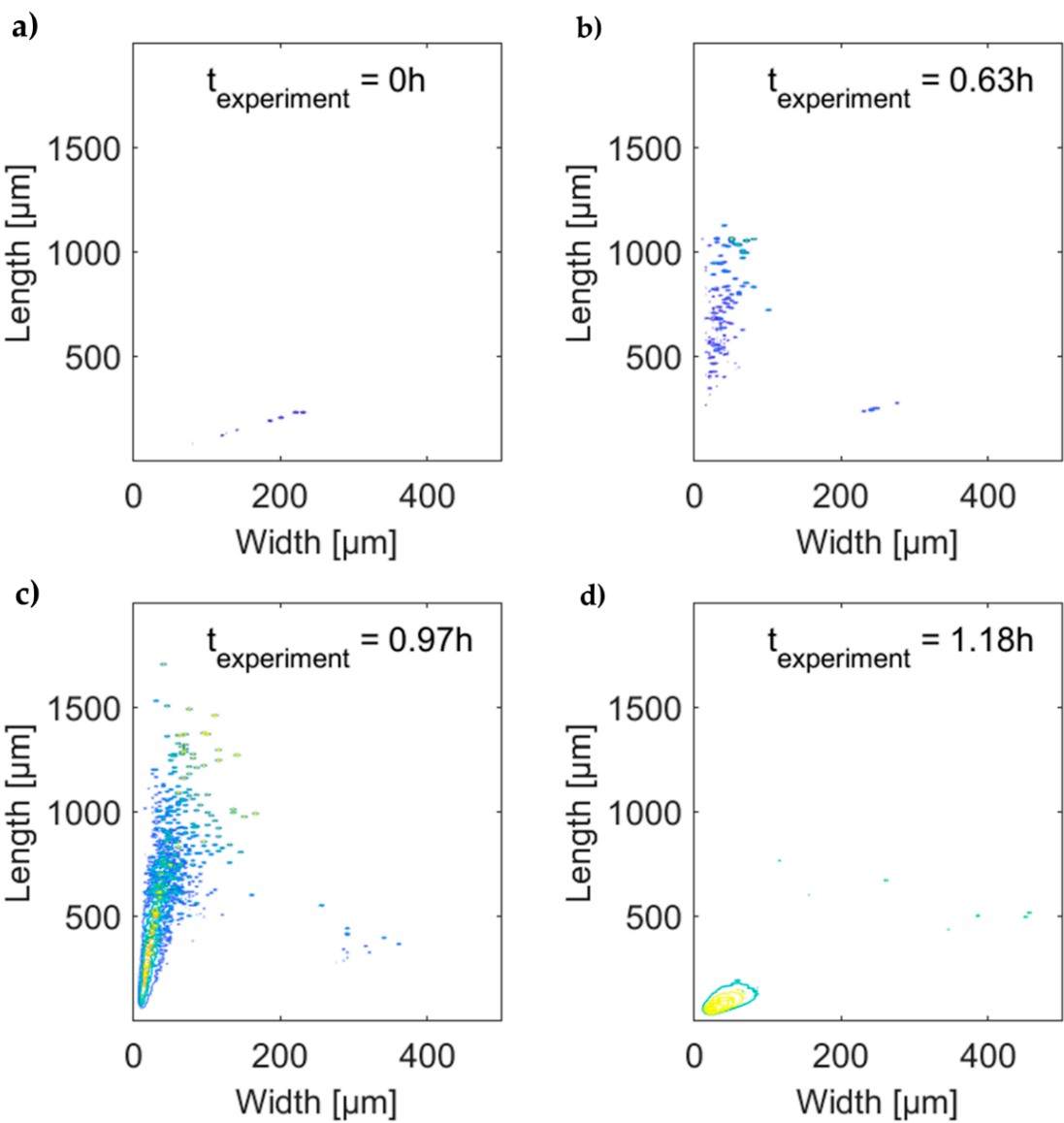

**Figure 14.** Length and width distributions of the needle-like crystals of thiamine hydrochloride monohydrate at different time points of Exp. 5, measured with the shadowgraphic probe. (**a**) measured crystals in the clear solution, (**b**) measurement of the first single needles, (**c**) significant number of crystals, (**d**) failed measurement as result of massive overlapping.

## 4. Conclusions

A novel inline shadowgraphic probe was utilized to determine the transient crystal size distribution in different crystallization processes, based on acquired greyscale images. For validation, three experiments in a well-known seeded $KH_2PO_4/H_2O$ cooling crystallization were carried out, and the crystal size distributions between the new shadowgraphic probe were compared with an established bypass online microscope and sieve analysis. The measured number and mass distributions showed a good agreement for both image-based techniques in all three experiments. The percentiles support the results, as they exhibit similar trends and values with an average deviation of 10–20 μm. Classifying effects, such as shifts in the crystal size distributions could not be observed up to a characteristic crystal size of 600 μm. This is confirmed by sieve analyses of suspension samples that were compared to the final mass-based distributions of the optical techniques. The experiments were performed for different seed loadings, and hence, different suspension densities up to 8%, and an optical density up to 11%, without any influence with respect to the measured distributions.

In addition, a fourth experiment, with the same substances at a higher starting saturation, temperature was executed. It was shown that the probe can be applied to these elevated temperatures and conditions, while a measurement with the bypass variant failed due to a blockage within the bypass tubes.

Needle-like thiamine hydrochloride monohydrate was crystallized from a clear thiamine/water solution when adding ethanol as an anti-solvent. The process was only investigated using the shadowgraphic probe, since measurements with the bypass online microscope failed due to blocking. In addition, the images captured by the online microscope had poor image quality, due to blurry edges of the imaged objects. The needle-like crystals could be measured in length and width up to a suspension density of three percent, until the image algorithm based on binarization failed, due to missing segmentation methods and massive particle overlapping. It was shown that the shadowgraphic probe can be applied to systems that form fragile crystals, where other techniques fail. It was found that the needle-like crystals align with the flow direction in the measurement gap, which offers a great potential for different image processing routines at a higher solid content.

A suspension density limit for the applied techniques cannot be generally determined. The measured optical density on pixel ratios does not necessarily match with the mass-based suspension density, but it can identify trends. The optical suspension density is mutually connected to the particulate state, such as size, distribution, and particle shape, and it must be evaluated for each system individually.

It was shown that the shadowgraphic probe is capable of monitoring the transient evolution of the PSD in a crystallization processes, with an extended range of operation conditions, and was compared to an established online bypass variant and sieve analysis. While bypass variants mainly suffer from blockage at high temperatures and supersaturations, the shadowgraphic probe can be applied under these conditions. In view of industrial application, it is desirable to extend the range of operation up to industrial conditions, i.e., suspension density, temperature, pressure, and chemical resistance. It is well known that image analysis fails at a high solid content, but mathematical algorithms (e.g., neuronal networks) have significantly developed in the recent years to overcome this gap. Hence, it is desirable to enhance the analysis range to industrially relevant conditions (e.g., larger suspension densities). With endoscopic probes, in combination with appropriate image analysis software, processes can be designed and scaled to industrially relevant size, as the development is less based on experience than on intrinsic data, including the particulate state.

In the state-of-the-art crystallization processes, the particulate phase is mostly not monitored and therefore largely unknown, which is one of the key problems in developing continuous crystallization processes.

**Author Contributions:** Conceptualization, E.T. and D.W.; methodology, E.T. and D.W.; investigation, formal analysis, validation, data curation and visualization, E.T. and D.W.; experiments, E.T., M.H. and D.W. writing—original draft preparation, E.T., D.W., M.H., H.L., A.S.-M., and H.-J.B.; writing—review and editing, D.W. and E.T., H.L., H.-J.B. and A.S.-M.; supervision, H.L., H.-J.B. and A.S.-M.; project administration and funding acquisition, H.-J.B. and A.S.-M. All authors have read and agreed to the published version of the manuscript.

**Funding:** This research received no external funding.

**Acknowledgments:** The support of Holger Eisenschmidt is gratefully acknowledged.

**Conflicts of Interest:** The authors declare no conflict of interest. The funders had no role in the design of the study; in the collection, analyses, or interpretation of data; in the writing of the manuscript, or in the decision to publish the results.

**Appendix A**

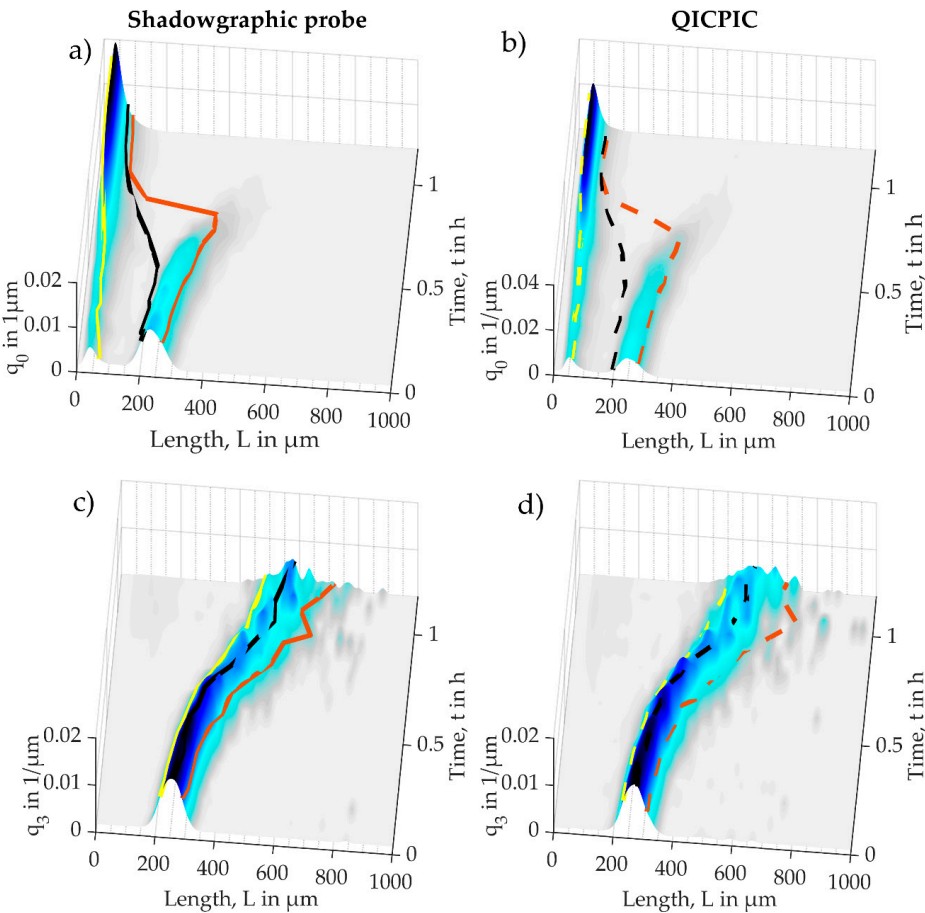

**Figure A1.** Exp. 1—KH$_2$PO$_4$ distributions (**a**) q$_0$-distribution shadowgraphic probe; (**b**) q$_0$-distribution QICPIC; (**c**) q$_3$-distribution shadowgraphic probe; (**d**) q$_3$-distribution QICPIC. *Solid lines*—percentiles of the shadowgraphic probe; *Dashed lines*—percentiles of the QICPIC; *yellow and red* −0.15 and 0.85 percentile distribution; *blue*—transient mean sizes of the distribution.

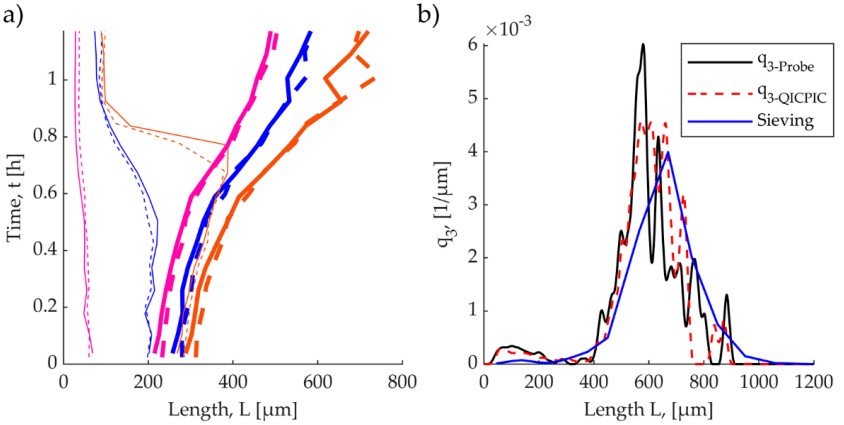

**Figure A2.** (**a**) Percentiles and mean values of the crystal size distributions for Exp. 1—KH$_2$PO$_4$ *thin lines*—percentiles of the q$_0$-distribution; *bold lines*—percentiles of the q$_3$-distributions; *Solid lines*—percentiles and mean values shadowgraphic probe; *Dashed* lines—percentiles and mean values QICPIC; *brown; magenta* −0.15 and 0.85 percentile of the corresponding distribution; *blue*—mean size of the corresponding distribution. (**b**) Comparison of the mass distribution: shadowgraphic probe, QICPIC, sieve analysis.

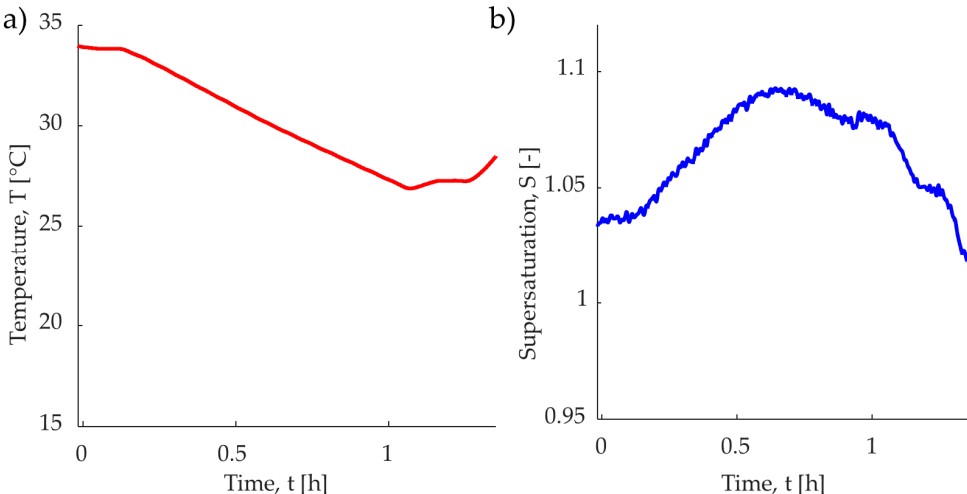

**Figure A3.** (**a**) Temperature profile of the Exp. 1—KH$_2$PO$_4$: saturation, seeding, linear cooling ramp, and final temperature. (**b**) Concentration profile of Exp. 3—KH$_2$PO$_4$: FTIR, offline samples and saturation curve.

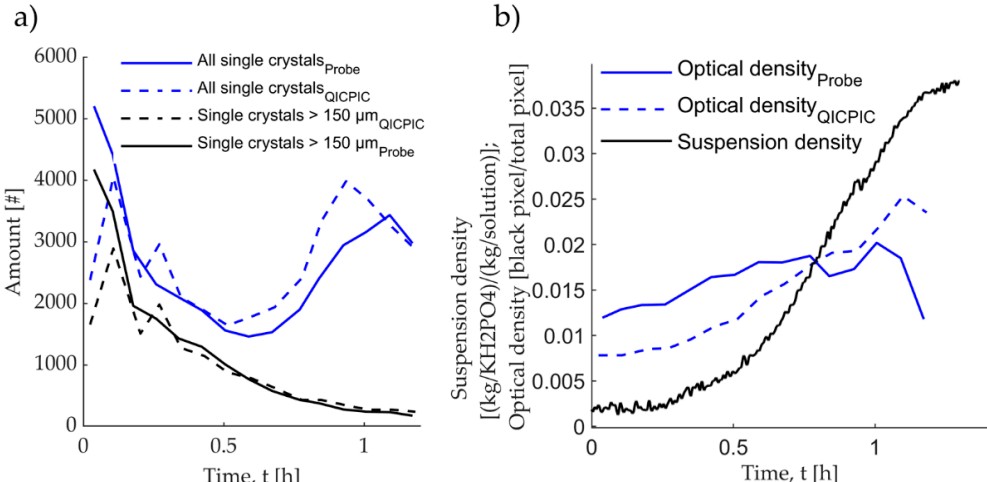

**Figure A4.** (**a**) Quantity of single crystals analyzed (**b**) the optical and suspension densities during experiment Exp. 1—KH$_2$PO$_4$.

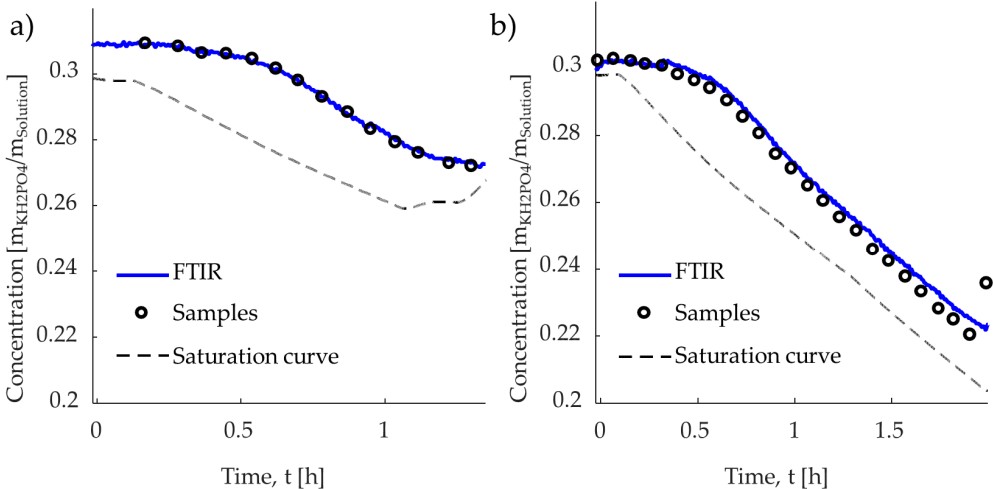

**Figure A5.** Concentration profile of (**a**) Exp. 1—KH$_2$PO$_4$ and (**b**) Exp. 2—KH$_2$PO$_4$; FTIR, offline samples and saturation Equation (1).

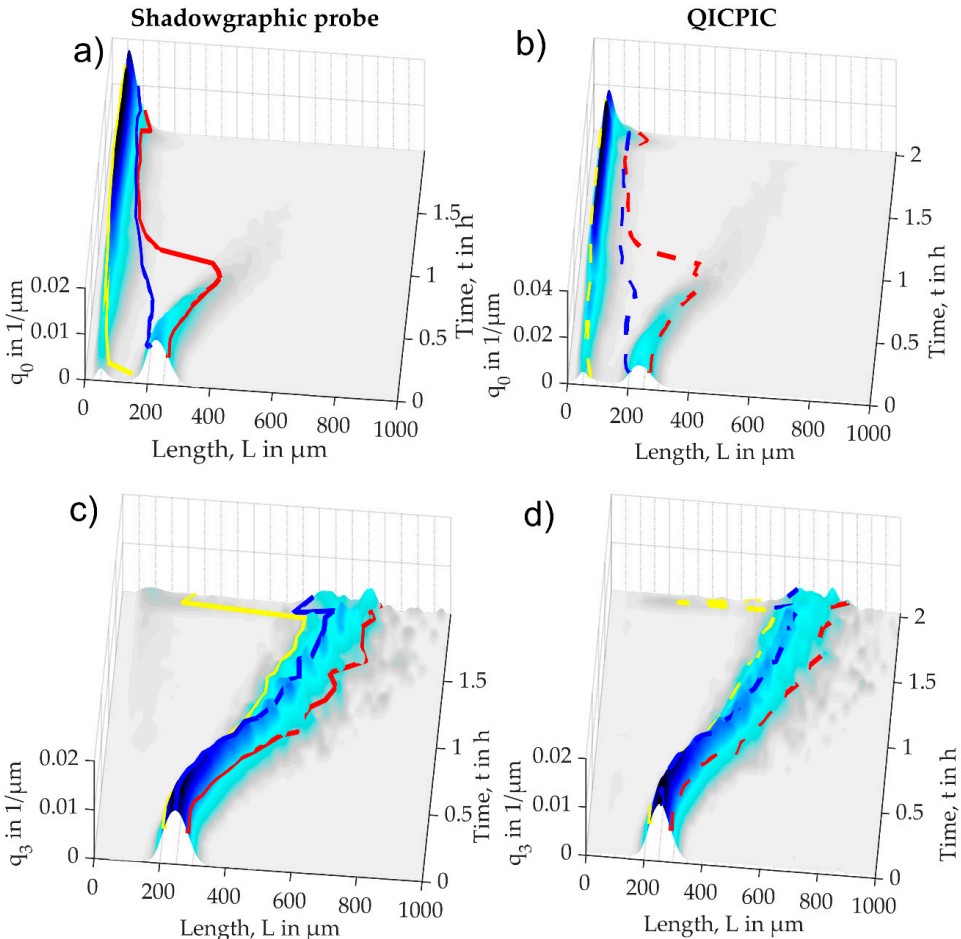

**Figure A6.** Exp. 2—KH$_2$PO$_4$ distributions (**a**) q$_0$-distribution shadowgraphic probe; (**b**) q$_0$-distribution QICPIC; (**c**) q$_3$-distribution shadowgraphic probe; (**d**) q$_3$-distribution QICPIC. *Solid lines*—percentiles of the shadowgraphic probe; *Dashed lines*—percentiles of the QICPIC; *yellow and red* −0.15 and 0.85 percentile distribution; *blue*—transient mean sizes of the distribution.

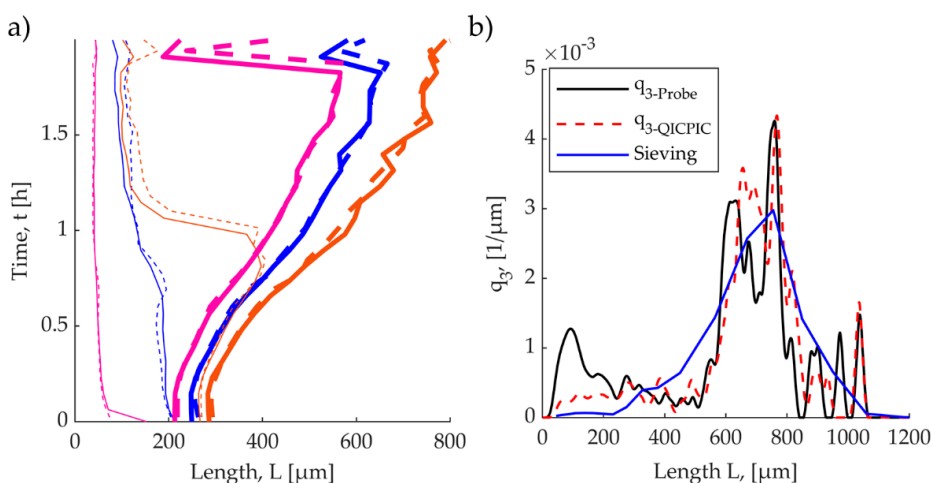

**Figure A7.** (**a**) Percentiles and mean values of the crystal size distributions for Exp. 2—KH$_2$PO$_4$ *thin lines*—percentiles of the q$_0$-distribution; *bold lines*—percentiles of the q$_3$-distributions; *Solid lines*—percentiles and mean values shadowgraphic probe; *Dashed* lines—percentiles and mean values QICPIC; *brown; magenta* −0.15 and 0.85 percentile of the corresponding distribution; *blue*—mean size of the corresponding distribution. (**b**) Comparison of the mass distribution: shadowgraphic probe, QICPIC, sieve analysis.

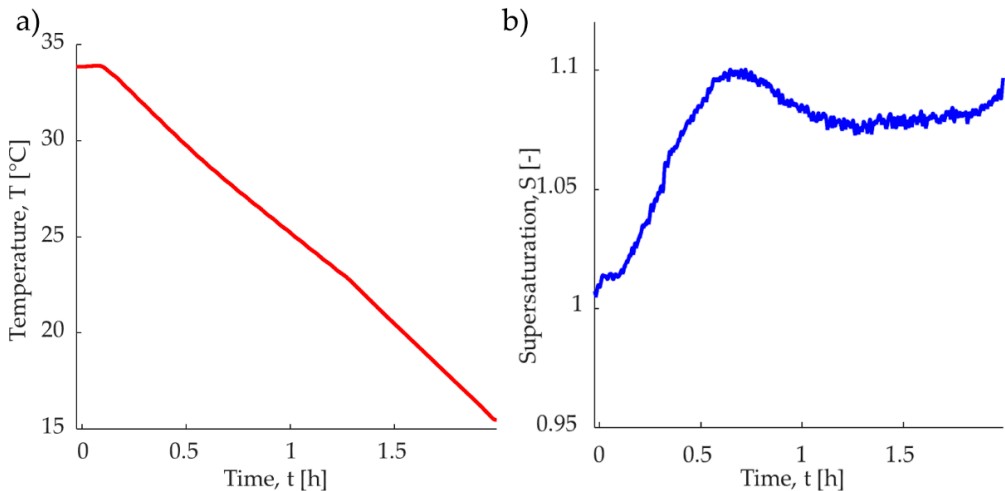

**Figure A8.** (**a**) Temperature profile of the Exp. 2—$KH_2PO_4$: saturation, seeding, linear cooling ramp and final temperature. (**b**) Concentration profile of Exp. 3—$KH_2PO_4$: FTIR, offline samples and saturation curve.

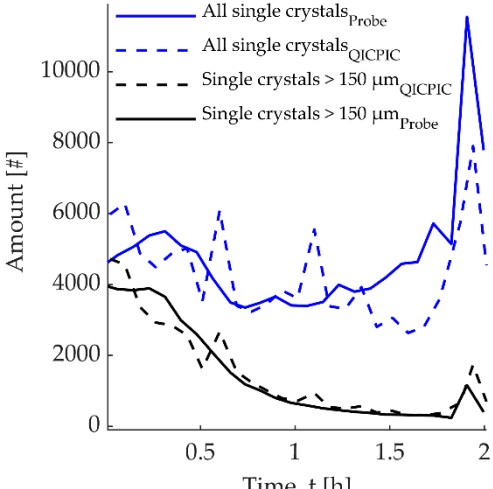

**Figure A9.** Quantity of single crystals analyzed during experiment Exp. 2—$KH_2PO_4$.

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
