# Peer review of "A Novel Shadowgraphic Inline Measurement Technique for Image-Based Crystal Size Distribution Analysis"

_crystals, doi:10.3390/cryst10090740_

Round 1

Reviewer 1 Report

The authors describe an alternative technique to measure crystal size and track crystal density within two different samples which is of interest in several different industries.   The shadow graphic probe outperforms QIPIC in high seeding volumes where the bypass line gets blocked which provides it with an added advantage of being able to detect needle like crystallisation processes much more confidently compared to QIPIC imaging technique. This new development is not only valuable in industry but also research application where we study crystallisation of proteins in fibril networks to help identify crystal density during data collection. Overall, this article is written well but there are areas which need grammatical attention and clarification.

General comments:

Comment 1, Introduction: The introduction demonstrates a gap in the current application process for sizing particle robustly during production and provides an overview of some of the technology applied for CSD but its not clear what the authors bring to the table. What is the goal of this manuscript? Is it to develop on new imaging technique which is more sensitive or just be able to detect higher density crystals or different morphologies? This should be stated in the first paragraph and then the aims should be stated again at the end of the introduction.

Comment 2:  Shadowgraphic optical probe and online microscope  section  explains the setup of the instrument and could be improved with the addition of a schematic showing the general setup, labelling the camera, probe, where the cell is normally positioned within the setup etc to show the difference in the two methods.

Comment 3: Line 142 ‘The techniques used in this article beside sieving are image-based techniques: an established bypass online microscope and a new telecentric shadowgraphic probe as described below.’ It is unclear what is meant by this sentence. Was a bypass online microscope and the telecentric shadow graphic probe incorporated into the new imaging based technique presented here? If so, you need to rephrase this sentence to reflect this.

Comment 4: Fig.1 shows the design of the DN50. Is this the authors own design or one already published, or a standard setup used in the field?? If its published references are required in the figure legend.

Comment 4: Experimental setup section has a nice diagram to depict the setup but no explanation that this is the setup that was used. An introduction sentence to the effect ‘ The setup of the system is shown in Fig.2’ at the start of your paragraph would help.

Comment 5: There is a mixture of past and present tense through the manuscript and sometimes within the same paragraph which should be fixed. There are also several grammatical errors that requires attention. Please re-read this this section and fix these errors throughout the manuscript.

Comment 6: line 331-337  In your crystal calculations overlapping crystals/aggregates were ignored. What percentage of the sample does this account for?

Fig 6 from which experiment was this sample taken from? The length axis may be better presented showing minor increments of 100 on the axis so the peak height can be seen clearly.

Comment 7: Line 435, ‘In summary, one can conclude that both optical measurement techniques are to evaluate the crystal size.’ This sentence doesn’t make sense. Do you mean both techniques are suitable to evaluate the crystal size? If so, this needs to be clarified.

Comment 8, line 448, The following comment was made in the manuscript ‘Only the larger crystal sizes seem to induce different measurement results. When comparing the initial and final crystal size (see Table 3), the inline probe measures larger particles while the measured initial crystal sizes agree well.’ While I agree that the different techniques for measuring the crystal size at the end stage of the process produced different results, I am not sure what is referred to in the second sentence. The inline probed agreed well at the start of the experiment with respect to the size but had the greatest difference to the sieve measurements in the end product and its values were actually smaller compared to the other two techniques.

Comment 9, line 451-457, which Figure is this referring too here? It would help the reader greatly if you would refer to the figures in your text to help follow your discussion more easily. This section needs clarity as its difficult to follow. I think what it is trying to explain why there is a difference in the inline probe results but it still not clear. The seems to be a 2-fold effect one is the secondary nucleation which alters the mean crystal size and the second is the agglomeration of the particles. Is this correct, if so, please clarify this paragraph. How is this overcome in the other 2 imaging techniques? Will this be a potential issue for samples that potentially higher secondary nucleation steps, or can this be controlled?

Comment 10, line 470, How large is the measurement window?

Comment 11: based on Figure 9, the 2 optical techniques although they show a similar trend, they show very different crystal amounts (>8000) in some points and density differences of 0.2. Are these within the acceptable errors?  What is the standard deviation for these results? Can you compare this to the standard deviations seen for suspension density values? Hence help quantify your statements in lines 492-500.

Specific comments:

Abstract: The last sentence of the abstract is not clear. This should be re-written.

Line 36, CSD needs to be defined as it’s the first time you mention it.

Line 76, ‘….summary of available techniques is given in the following.’ Is missing something at the end of the sentence. Add ‘…following paragraphs’ to complete the sentence.

Line 77, this sentence doesn’t make sense. It needs to be re-worded.

Line 70, ‘In contrast to the broad range of applications discussed the following focus will be on crystallization.’ Should be changed to ‘In contrast to the broad range of applications discussed the focus in this article will be on crystallization.

line 126, ‘…various authors have used this…….’ Should be re-written as ‘….various publications demonstrate the use of this….’

Line 131 ‘…are inserted in..’ should be ‘….are inserted into the…’

Line 133, focus plane should be ‘focal plane’

Line 139, ‘This measurement principle is completely noninvasive and was developed in human medicine but found nowadays also application in process engineering.’ This sentence is clumsy and needs re-writing.

Line 149-154 – poor grammar and needs to be re-written.

Line 173, ‘Further, the depicted particles captured by the camera appear smaller the larger the distance is from the  entocentric lens…’ change to  ‘Further, the depicted particles captured by the camera appear smaller, the larger the distance it is from the entocentric lens ..’

Figure 2- The figure legend needs more explanation of what the diagram is showing. The abbreviations in the figure needs to be spelt out in the figure legend. What is M, QICPIC, FTIR, PC.

Line 200, ‘…… and a PT100 is used for temperature monitoring.’ Should be change to ‘….and a PT100 is used to monitor the temperature.’

Line 208, flow rate flow (≈ 40 L/h) delete the second ‘flow’

Line 208’……. All bypass tubes are temperature-controlled by a thermostat and double jacketed tubes, which..’ should be change to ‘…All bypass tubes are double jacketed and are temperature-controlled by a thermostat which was..’

Line 222, ‘…hydrochloride monohydrate in a needle-like shape, depicted in Fig. 3b changed to ‘…hydrochloride monohydrate grew needle-like shape, depicted in Fig. 3b.’

Line 247, ‘Table 2 depicts the conditions of the single thiamine hydrochloride and four KH2PO4 cooling 247 crystallizations.’ This sentence doesn’t make sense. Please clarify.

Table 3, what is B1?

Equation 2, what is C, is this concentration? Ensure all symbols are defined in the text for all equations.

Line 298 diagramm spelt incorrect. Should be diagram.

Line 397, It is stated ‘The experiment was carried out without 307 temperature control at 25 °C.’ This is a confusing statement. Was there temperature control or not or was it just set to 25°C and not maintained at this temp. Please clarify.

Figure 10 densitys spelt incorrect. Should be densities.

Fig 12 interesting shows a different density of crystals in the image. Can you explain this? Are these images taken at the same time point in the reaction?

Author Response

Dear Reviewer,

At first thank you for your time and your intense review to our work, we appreciate! We reworked our script and tried to include your thoughts and comments. I hope I got you right and changed the manuscript in all points towards your input. Otherwise please leave me comments, thank you in forward.

Reviewer 2 Report

There is a real need for more experimental inline methods for the determination of crystal size distributions (CSDs) during crystallization. The shadowgraphic method reported in this paper has advantages over other available methods, and thus potential for much wider use. The clear and detailed write-up in this manuscript should facilitate understanding of the method and adoption by other researchers. This is a really well-written manuscript that combines a good review of the field with an in-depth description of the apparatus and the experimental method, followed by comparison with results from alternative/conventional methods for measuring CSDs, and a discussion that brings out the advantages of the technique. For the systems investiated here the methods works well. I therefore strongly recommend that it is published, to stimulate more use by the community and further critical evaluation with other crystallisation systems, so that we can get a better understanding of whether the results presented here can be generalised.

Author Response

Dear Reviewer 2,

thank you very much for your time. we appreciate. We made some minor changes towards english and made some improvements. We hope you like the result and share your oppinion that it stimulate further work.

Best regards